# Unlaid Eggs: Ovarian Damage after Low-Dose Radiation

**DOI:** 10.3390/cells11071219

**Published:** 2022-04-04

**Authors:** Elisabeth Reiser, Maria Victoria Bazzano, Maria Emilia Solano, Johannes Haybaeck, Christoph Schatz, Julian Mangesius, Ute Ganswindt, Bettina Toth

**Affiliations:** 1Gynecological Endocrinology and Reproductive Medicine, Medical University of Innsbruck, 6020 Innsbruck, Austria; elisabeth.reiser@i-med.ac.at; 2Laboratory for Translational Perinatology-Focus, Immunology, Chair of Obstetrics and Gynecology, Focus, Obstetrics, University Hospital Regensburg-St. Hedwig Clinic, 93053 Regensburg, Germany; victoria.bazzano@ukr.de (M.V.B.); maria-emilia.solano@ukr.de (M.E.S.); 3Institute of Pathology, Neuropathology and Molecular Pathology, Medical University of Innsbruck, 6020 Innsbruck, Austria; johannes.haybaeck@i-med.ac.at (J.H.); christoph.schatz@i-med.ac.at (C.S.); 4Diagnostic & Research Center for Molecular BioMedicine, Institute of Pathology, Medical University of Graz, 8010 Graz, Austria; 5Department of Radiation Oncology, Medical University of Innsbruck, 6020 Innsbruck, Austria; julian.mangesius@i-med.ac.at (J.M.); ute.ganswindt@i-med.ac.at (U.G.)

**Keywords:** low-dose radiation, ovarian damage, fertility preservation, mouse model, follicle count, oocyte

## Abstract

The total body irradiation of lymphomas and co-irradiation in the treatment of adjacent solid tumors can lead to a reduced ovarian function, premature ovarian insufficiency, and menopause. A small number of studies has assessed the radiation-induced damage of primordial follicles in animal models and humans. Studies are emerging that evaluate radiation-induced damage to the surrounding ovarian tissue including stromal and immune cells. We reviewed basic laboratory work to assess the current state of knowledge and to establish an experimental setting for further studies in animals and humans. The experimental approaches were mostly performed using mouse models. Most studies relied on single doses as high as 1 Gy, which is considered to cause severe damage to the ovary. Changes in the ovarian reserve were related to the primordial follicle count, providing reproducible evidence that radiation with 1 Gy leads to a significant depletion. Radiation with 0.1 Gy mostly did not show an effect on the primordial follicles. Fewer data exist on the effects of radiation on the ovarian microenvironment including theca-interstitial, immune, endothelial, and smooth muscle cells. We concluded that a mouse model would provide the most reliable model to study the effects of low-dose radiation. Furthermore, both immunohistochemistry and fluorescence-activated cell sorting (FACS) analyses were valuable to analyze not only the germ cells but also the ovarian microenvironment.

## 1. Introduction

The overall five-year relative survival rate of all childhood cancers has significantly improved over the last thirty years from 58% to 83%, giving rise to a special population: adult survivors of childhood cancer after cytotoxic chemo- and radiotherapy (RT) with the desire to have children regardless of their medical history [1,2]. Fertile women have lower chances to become a mother after surviving cancer treatments [3].

As chemotherapy—in contrast to radiation—is mainly applied to all cancer patients, a myriad of studies already exists, which indicate that commonly used agents cause premature ovarian insufficiency (POI) by inducing the death or accelerated loss of primordial follicles as well as damage to blood vessels, stromal cells, and immune components [4,5,6,7,8,9,10,11,12]. However, data on (low-dose) radiation in fertile women are urgently needed.

Radiotherapy accelerates oocyte depletion by different mechanisms such as apoptosis or oxidative stress, leading to acute ovarian failure (AOF), POI, and menopause. Radiotherapy is a cornerstone of state-of-the-art cancer therapy in children, women, and men. At present, around 50–60% of all patients with long-term cancer (45–50% curative success for all cancer types) receive radiation therapy alone or in combined treatment schedules [13]. The direct and undisputed beneficial effects of targeted radiation on tumor cells result from preventing further cell proliferation or inducing cell death. Irrespective of this, the surrounding organs can be directly damaged by scattered radiation [13,14].

The human ovary contains a finite pool of primordial follicles (PMF), which comprise the ovarian reserve. Its maximum is already reached as a fetus at five months of gestation, which then declines with increasing age and culminates in the menopause at an average age of 50–52 years [14,15]. This fixed number of oocytes is non-renewable and must provide for the entire reproduction cycle throughout adult life. The maximum reserve during fetal life is followed by atresia and the loss of more than half of the originally developed germ cell pool at the time of birth. Only oocytes that are enclosed by a sufficient number of epithelial and stromal cells are able to survive. Therefore, the interaction between oocytes and stromal cells is of utmost importance even during prenatal development [16,17,18]. Different cell populations of the ovarian cortex can be defined by gene expression analyses: stroma (83%); oocytes (0.2%); perivascular (10%); endothelial (5%); granulosa (1.2%); and theca and immune cells (0.4%) [19,20].

In the last decade, maintaining the gonadal function and preserving fertility after successful cancer treatment have become critical and have increased the concerns of young fertile patients. The options depend upon the age of the patient, their physical state, the administered agent, and the start of the cancer treatment. Regarding radiation of the pelvis, ovaries can be transposed outside the radiation field (ovarian transposition). This surgical technique has been performed since 1952 with low success rates and a high risk of POI (33–100%) also due to scattered radiation [21,22,23]. Patients undergoing brachytherapy seem to benefit most with ovarian survival rates from 77.8–100% [24]. However, an increased risk of ovarian cysts after ovarian transposition has been reported, ranging between 0 and 34% [25,26]. Together with other complications such as abdominal pain, hematomas, tubal ligation, ischemia, or small bowel obstructions, reoperation was necessary in 34.7% of patients with complications [24]. Therefore, ovarian transposition may be offered to women with planned pelvic radiation without chemotherapy as being recommended by international guidelines [27,28,29,30]. Of note, supporting evidence is weak; therefore, ovarian transposition may not act as a safe procedure for fertility preservation with special regard to the above-mentioned complications. Another technique—which can only be offered after puberty—is the cryopreservation of oocytes before the legal age and the cryopreservation of (non) fertilized oocytes thereafter. In order to cryopreserve, women need to undergo controlled hormonal stimulation before the oocytes are transvaginally retrieved [31]. Therefore, this technique can only be offered if the cancer treatment can be postponed for at least 2–3 weeks.

The cryopreservation of ovarian tissue was first performed by Prof. Dr. Donnez in 1997. After successful cancer treatments, the tissue can be thawed and transplanted onto the remaining ovary or into a peritoneal tissue pouch near the ovaries. As the first successful ovarian tissue transplantation was performed in 2004, this technique is now routinely performed. The resumption of cyclic hormone production can be achieved in up to 63% of cases, and live birth rates (LBR) are described as 23% per transplantation [32,33]. To date, more than 170 live births after the transplantation of frozen–thawed ovarian cortical pieces have been reported, and the transplanted ovarian tissue remains active from six months to three years [32]. However, the cryopreservation of ovarian tissue is not established in children and only parts of the ovary (40–50%) can be cryopreserved whereas the major part stays within the pelvis and is exposed to radio-chemotherapy.

Although surgery and chemotherapy cause several known severe effects on the gonads [4,5,6,7,8,9,10,11,12], we critically evaluated the current data on radiation-induced damage in ovarian tissues, which was formerly neglected or stepped back in studies focusing on fertility preservation. The latest reviews in the field of radiation-induced ovarian damage focus mainly on the genetic effects and possible DNA damage [34,35]. Our aim was to set up an experimental setting for animal and human models and allow international interdisciplinary investigations in order to provide new therapeutic options, especially for young cancer patients. Special attention is drawn to low-dose radiation (LDR) as this affects a large number of female cancer patients and only little knowledge exists.

## 2. Materials and Methods

We searched the PubMed, Cochrane, Embase, Bio-SISS, and Web of Science databases containing the keywords low-dose radiation, ovarian damage, fertility preservation, young cancer patients, mouse model, stroma, immune cells, oocyte, and follicle count in English language literature from 1980 to January 2022 (Figure 1). The inclusion criteria consisted of original peer-reviewed data, mice models, and low-dose radiation in fertile women; the exclusion criteria were reviews, genetic effects (germ cell mutations) after environmental radiation (e.g., radiofrequency electromagnetic fields), and population studies of atomic bomb survivors or after radioactive accidents as well as animal models other than mice.

With regard to the animal models included in Table 1, we revised the experimental work that included the use of mice exposed to low-dose radiation (<1 Gy) applied in vivo. Here, we prioritized in vivo radiation that we considered to be more comparable with the clinical treatments than the in vitro approaches. Remarkably, the in vitro approaches showed consistent deleterious effects of LDR and were instrumental in elucidating the molecular and cellular pathways of tissue damage in the gonads. However, our interest was to discuss the work that mimicked female survivors of childhood and postpubertal cancer, which constituted a population of high clinical relevance. In vitro, the results became independent of the factors external to the gonads such as the influence of hormones, immune reactions, and general health status that may play an important role in cancer patients. For these reasons, we focused on in vivo studies to evaluate the progress and limitations of the current experimental models. Finally, additional inclusion criteria were that radiation was carried out only on mice at postpartal day 5 or older. After postpartal day 5, the germ cell cyst breakdown and primordial follicle assembly is finalized in the mouse ovary, mirroring the postnatal stages of ovarian development in humans (for example, in patients suffering from cancer in childhood). We did not include abstracts or conference proceedings. The selection of original work was undertaken by following the PRISMA guidelines [36].

## 3. The Human Ovary and Folliculogenesis

### Folliculogenesis

Human folliculogenesis has been described in detail in the literature [45,46]. Meiosis of the oogonia starts in early fetal life and the final PMF pool is formed around 24 gestational weeks in humans and around birth in rodents (Figure 2). Once meiosis is initiated, mitosis ends; therefore, the individual PMF pool is already fixed before birth. Of note, POI occurs in 1% of women under the age of 45 years without the influence of gonadotoxic treatment due to a low PMF pool [47]. After menarche, the follicles either start to grow or become atretic. The activation of PMF includes different pathways such as PI3K/PTEN7Akt and Hippo [48,49,50,51]. However, the detailed mechanisms of the regulation of the PMF pool over the reproductive female lifespan remain unknown. PMFs are mainly located in the ovarian cortex, which represents the poorest vascularized zone in the human ovary. Both primordial and early growing follicles are dependent on stromal vessels as they do not rely on an independent vascular network [52]. PMFs consist of an oocyte surrounded by granulosa cells [53]. The high proliferation rate of granulosa cells explains their sensitivity to chemotherapy and radiation. The secondary follicle is characterized by the formation of a zona pellucida, stromal cells, and an undifferentiated theca layer. After differentiation into a theca interna and a theca externa layer, the stage of a pre-antral follicle is reached followed by an early antral follicle with fluid-filled cavities and a large antral follicle with a visible large antrum and a marginal oocyte [46,53].

## 4. Application of Low-Dose Radiation in Clinical Practice

Cancer treatment can cause female infertility in different ways. Fertility impairment as well as an elevated risk of pregnancy complications primarily depend on the applied radiation dose and on the composition of a systemic co-treatment. Thereby, an additive effect is likely for combined chemoradiation schedules. In addition, the age at treatment and the location of the treatment target are decisive factors [63].

Pediatric cancer survivors are known to exhibit a significant increase in the risk of infertility at doses ranging from 1 to 5 Gray (Gy) to the uterine region (RR = 1.33), which is further elevated at >20 Gy (RR = 2.50) [64]. The increasing use of intensity-modulated radiation therapy (IMRT) and volumetric modulated arc therapy (VMAT) techniques can cause higher co-radiation doses (radiation leakage) deposited in the uterine and ovarian region compared with standard conformal RT [65]. Modern radiation planning techniques with ovary-sparing protocols can also be applied to significantly reduce ovarian radiation exposure [66]. However, the proof of functional outcome advantages using this method is still lacking. Moreover, position variability as well as ovarian movements complicate ovarian protection using IMRT. The effect of co-radiation ranging from the lowest mGy doses up to 0.5–1.0 Gy as a consequence of more distant target sites or optimized organs at risk-sparing during treatment is still speculative or a matter of linear extrapolation (linear non-threshold (LNT) hypothesis) [67]. In addition, individual and organ-specific differences in the response to LDR as well as cell–biological mechanisms such as the bystander effect further complicate a generally valuable risk prediction [68]. Nevertheless, for a few very low doses, the overall risk of a certain health effect may be clinically identical with the absence of a radiation-induced excess risk. This might also be valuable for uncertain risk estimations of infertility or germ cell mutations following curative radiotherapy with a low-dose involvement of the reproductive tract.

In order to prove the presence, extent, or absence of such risks for female cancer patients at a certain assumptive threshold, the only available data are provided by epidemiological studies, which principally employ an observational, non-experimental approach. However, data from epidemiological studies are generated by the uncontrolled conditions of everyday life and randomized controlled trials are unacceptable for the investigation of actual or potential hazardous exposures [69,70,71]. Thus, radiobiological clues to investigate the low-dose effects of high-energy radiation first have to be obtained from dose-relationship studies using in vitro and in vivo animal models [72].

The dose–response relationship describing the excess risk of stochastic health effects (radiation-induced cancer and germ cell mutations) following low-dose levels of exposure to ionizing radiation is more and more controversially discussed. The standard approach for the purposes of radiological protection is based on the hypothesis that radiation-induced risks are directly proportional to the administered dose, as described by the LNT hypothesis. Nevertheless, several radiobiologists have argued that this approach underestimates the current risks (i.e., the relationship is properly described by dose–response curves of a supralinear shape) or that there is a threshold dose below which either no effect or even a beneficial (hormetic) effect is likely to exist [73].

In the past, the International Commission on Radiological Protection (ICRP) released several reports concluding that a low dose threshold seems to exist for radiation-induced malignancies of certain tissues as well as for infertility or germ cell mutations [74,75,76]. In their opinion, this evidence does not favor the existence of a universal threshold. The LNT hypothesis, combined with an uncertain dose and dose-rate effectiveness factor (DDREF) for extrapolation from high doses, still represents the propagated basis for radiation protection at low doses and low-dose rates by the ICRP, World Health Organization (WHO), and other official authorities.

However, the almost dogmatic LNT approach is being increasingly questioned within the expert community due to the continuously increasing amount of reports that prove hormesis, an adaptive response, and individually varying susceptibility to high-energy low-dose and/or low-dose-rate radiation [77]. With regard to the risk of infertility after chemo-co-radiation, it is of utmost relevance to gain further experimental insight from suitable translational research and bias-revised epidemiological studies in order to investigate the potential existence of lower and upper threshold doses in gonadotoxicity. Both the evidenced presence of threshold doses and a finally confirmed linear dose relationship from below 0.5 to 1 Gy would decisively improve the process of therapy decision making for fertile female cancer patients with a good survival prognosis.

## 5. Animal Models and LDR: Effects on the Ovary

### 5.1. Selection of Model

The implementation of animal models in studies aiming to understand the effects of ionizing radiation on female and male reproductive health has a long history [78]. Recently, the possibility of replacing high-dose radiation with LDR as a safer alternative [42] made the development of new models imperative to investigate this phenomenon in depth. To date, mouse models are the most widely used in radiation experiments although the effects of LDR on rats [79], the nematode *C. elegans* [80], and adult female rhesus macaques [81] have also been reported. Mouse models are favored over other animal models because mouse ovaries contain similar cohorts of follicles during the continuum of follicular development with similar functions (autocrine, paracrine, and endocrine functions as well as the production of a mature oocyte). Moreover, a variety of genetically modified strains as well as a broad set of tools and detection methods for analyzing the potential pathways involved in tissue damage upon radiation are available in mouse models. These models may be instrumental in studying the role of immunological processes in tissue damage and regeneration because immune responses in mice are well-studied and resemble human immunity more closely than other animal models.

Based on this knowledge, in this section we review the publications of the effect of LDR on the ovaries, particularly in mouse models, in order to unveil current shortcomings and possible avenues for future research. Of note, we revised the experimental work that included the use of mice exposed to low-dose radiation (<1 Gy) applied in vivo, as mentioned in the inclusion and exclusion criteria. We chose in vivo radiation as it is more comparable to the situation in humans than the in vitro approaches [82]. Remarkably, the in vitro approaches showed consistent deleterious effects of LDR and elucidated the molecular and cellular pathways of tissue damage in the gonads [54,55,82].

### 5.2. Mouse Model: Current Knowledge and Further Approaches

As depicted in Table 1, multiple protocols have been applied to mice with regard to the source (γ-ray and X-ray), rate (0.037 Gy/s for 27 s to 2.1 Gy/min for 0.48 min), total dose (0.45 Gy–1 Gy), and body area exposed to the irradiation applied (targeted versus TBI). These experimental studies contemplated additional variables that could influence the gonadotoxic effects including the genetic background of the mouse strains (C57BL/6, BALB/c, CD1, NMRI, 129S2/SvPasCrl) and the age of the mouse at the time of radiation (5 d to 6 w). These observations mirror the research in women in which the success of fertility preservation also depended on the age and physical state of the patient.

The models used γ-ray radiation more frequently [37,38,39,40,41,42,43], which is in line with current medical practice that favors the use of γ radiation in humans. Only two studies employed X-ray radiation [40,44] and showed a similar impact on the ovarian reserve as an equal dose of γ irradiation, supporting the further use of γ-ray radiation in the experimental setting.

With regard to the radiation dose, diverging outcomes were reported for the very low radiation dose range (0.02–0.1 Gy); a few studies observed no differences in the ovarian reserve compared with the non-radiated controls [37,42] and others reported a small [40] or near-complete obliteration of the primordial follicles [43]. These contrasting observations were not dependent on the mouse strain or age at radiation [42,44], opening questions about which additional environmental factors may influence the effects.

Within the LDR range of 0.45–1 Gy, a decrease in the number of germ cells was unanimously reported in the work reviewed [37,38,39,40,41,42,43,44]. The mainly applied 1 Gy dose generated deeper damage to the ovary than the lower doses, indicating that the severity of tissue damage is dose-dependent [41,42]. Furthermore, the response of the follicles to radiation depended not only on the radiation dose but also on the stage of the follicular development. Early follicles from the primordial to secondary stages appeared to be more radiosensitive compared with the large antral follicles [41,42,44]. The mechanisms of the LDR-induced depletion of the primordial follicles require the expression of the p63 gene, a central tumor suppressor in mammals [37], and included a rapid and massive wave of apoptosis of quiescent oocytes involving caspase-2-dependent activation followed by the activation of caspase-3 and -9 [83].

Importantly, the effect of 0.45–1 Gy of LDR remained consistent throughout all developmental ages investigated. Of note, all studies included young mice of the prepubertal period [37,38,40] (comparable with patients of childhood cancer) as well as peripubertal 6-week-old mice (comparable with peripubertal young cancer patients) [83]. Hence, the range of life stages considered in the mouse studies should be expanded to higher ages (e.g., 3–5 months old C57BL/6J mice representative of humans between 20 and 30 years) to provide a more complete spectrum of ovarian developmental stages and also reflect adult women undergoing radiation. Remarkably, the studies with prenatal and neonatal radiation exposure showed that ovaries from younger mice were more vulnerable to tissue damage [83,84]. These observations suggest that once the germ cells undergo the primordial follicle assembly, their vulnerability to radiation is less dependent on the age of the female.

As mentioned, differences were low with regard to the mouse strains. Although the expression of osteopontin (Spp1)—a gene involved in wound healing, inflammation, and fibrosis—was significantly reduced in BALB/c mice in response to radiation compared with 129S2/SvPasCrl (129) strains, no differences in the depletion of the ovarian reserve occurred [41]. These results suggest that, in humans, the impact of the genetic background (e.g., the ethnicity) on the effect of radiation on the ovarian reserve may be low although epidemiological studies are needed to further assert this assumption.

The pretreatment of mice with sphingosine-1-phosphate (S1P), an antagonist and negative regulator of apoptosis, seems to be a promising ovarian protection strategy. The administration of S1P 2 h before radiation protected the ovarian reserve and irradiated mice were able to produce live offspring [43]. These observations were subsequently confirmed in a rhesus macaque model [81], raising questions about the feasibility of intra-ovarian S1P administration prior to radiation therapy in routine clinical settings for humans.

The location of the radiation field may also influence the detrimental effects on the ovaries. In line with this, most mouse studies utilized TBI (Table 1). Recently, the development of microirradiator platforms allowed the mimicking in mice of targeted radiation that is used in clinical settings that does not require TBI. Using this technology, a similar extent in the reduction of the primordial follicles was observed when radiation was applied to the total body or targeted one (T1) or two (T2) ovaries of the mouse [44]. The T1 group showed significant decreased primordial and growing follicles compared with the non-targeted contralateral ovaries [44]. These results support an advantage in the use of targeted irradiation to protect normal tissue function from off-target radiation damage in mice as it has been proposed in women [85]. Nevertheless, it remains to be investigated whether the off-target effects of radiation on the contralateral ovary or unpredicted compensatory effects could occur in the long term.

In conclusion, the current mice studies showed that both X- and γ-ray radiation led to similar ovarian damage. Radiation-induced ovarian damage was consistent in LDR between 0.45–1 Gy and the effects were independent of the strain or age of the mice. More variability was observed in the case of lower radiation doses employing between 0.02–0.1 Gy, which may offer possibilities to preserve the ovarian reserve, at least partially. However, the contrasting observations reported did not seem to relate to the age or genetic background of the mice, suggesting that additional environmental factors (e.g., diet or microbiome) might be at play. Such factors require urgent identification in order to establish a reliable method for fertility preservation in young cancer patients.

### 5.3. Effects of Low-Dose Radiation on Ovarian Stromal and Immune Cells

In contrast to the abundant research on radiation-induced cell death in oocytes, few studies evaluated the effect of LDR exposure on the ovarian stroma and microenvironment in general. Intriguingly, in humans exposed to radiation treatment, fibrosis was observed in a myriad of tissues including lung, breast, heart, and intestine [86,87] although information about fibrosis in the ovary is still rare and requires further investigation. To our knowledge, no data exist on the FACS analysis of human ovarian tissue after radiation.

Detected immune cells in the human ovary include macrophages, dendritic cells, neutrophils, eosinophils, mast cells, B lymphocytes, T lymphocytes, natural killer cells, toll-like receptors, and macrophages [20,88,89,90]. Their amount varies throughout the menstrual cycle. Macrophages seem to modulate the ovarian function by the regulation of ovulation [91]. NK cells comprise 10–15% of lymphocytes and are well-studied in the endometrium [92,93,94,95].

The current evidence indicates that despite follicle depletion, LDR neither triggered the formation of fibrosis in the ovarian stroma of mice of diverse strains [41,42] nor affected the ovarian vasculature [42]. Of note, high-dose radiation (15 Gy) in rhesus macaques led to fibrosis 10 months after radiation, measured by collagen accumulation [96]. Amargant et al. compared fibrosis in ovarian stroma after radiation dependent on a treatment with an antiapoptotic agent (sphingosine-1-phosphate (S1P)). However, S1P did not protect the ovarian stroma from fibrosis after exposure to high-dose radiation; whether S1P can overcome this effect following low-dose radiation remains to be studied. High-dose radiation (15 Gy) in rhesus macaques (n = 3) led to elevated interleukin 6 (IL-6) and monocyte chemoattractant protein-1 (MCP1) serum levels 10 months after the treatment. Moreover, macrophages were present in the ovarian cortex of the treated animals [97].

Other possibilities for measuring radiation-induced damage include γ-H2AX. Histone yH2AX plays a role in the DNA damage response of double-strand breaks (DSB). Its value has been correlated with DNA damage and repair in a variety of cell types and tissues, especially in peripheral blood [98,99]. Radiation-induced γ-H2AX levels in peripheral blood rapidly increased within 30 min and reached a maximum by 1 h dose-dependently in a study by Lee et al. [100]. In mouse models, γ-H2AX was detected in follicular somatic cells and oocytes in irradiated ovaries as a sign of DNA damage [37,40]. The role of γ-H2AX in DNA repair in cultures of non-dividing primary human fibroblasts after very low-dose X-ray radiation (1 mGy) was studied by Rothkamm et al. [101]. The group found that DSBs remained unrepaired for several days whereas efficient DNA repair was observed after high-dose radiation, suggesting that the cellular response to DSBs is different for low and high radiation doses [102]. In organoids, a less efficient activation of DNA damage response was observed at 0.25 Gy than at 1 Gy [103].

Emerging techniques such as proton RT may enable the better sparing of ovarian tissue and preserve fertility and endocrine functions such as protection using IMRT, especially for targets close to the ovaries. However, these experimental techniques will not be available for the vast majority of cancer patients in the foreseeable future.

## 6. Conclusions

The group of adult survivors of childhood cancer after cytotoxic chemo- and radiotherapy having the desire to have children regardless of their medical history is constantly rising. This review revealed a lack of knowledge about LDR applied to the ovary. The existing data almost solely focus on the impact of LDR on the follicle count. We urgently recommend further interdisciplinary research utilizing mouse models and combining immunohistochemistry and FACS analyses to establish an overall picture of the effects of LDR on female fertility with special regard to young cancer patients.

## Figures and Tables

**Figure 1 cells-11-01219-f001:**
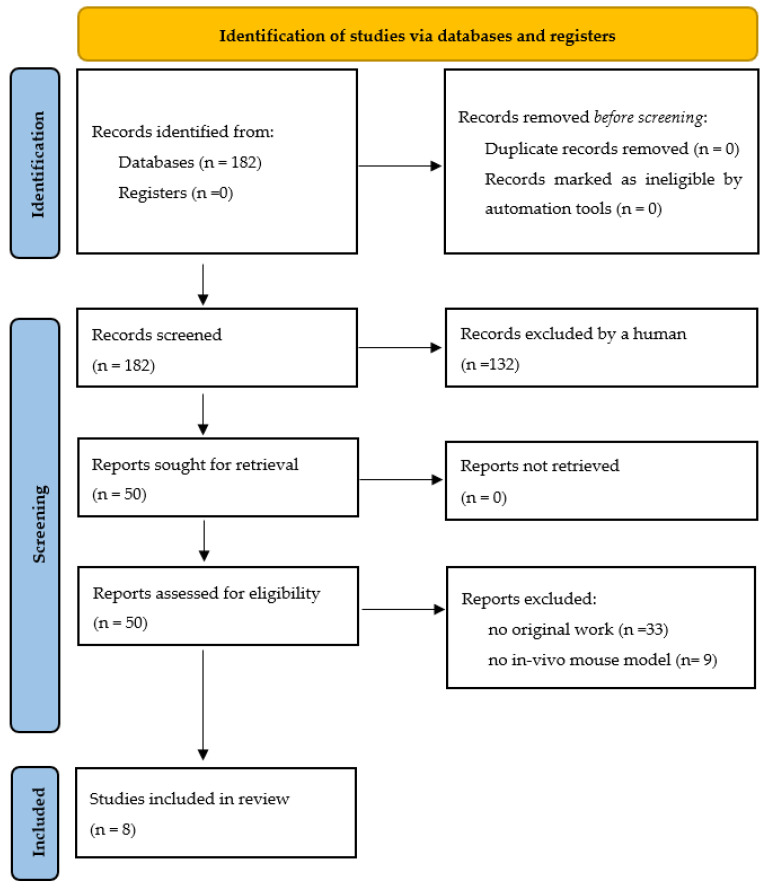
Flow diagram displaying the included searches of databases concerning the mouse model.

**Figure 2 cells-11-01219-f002:**
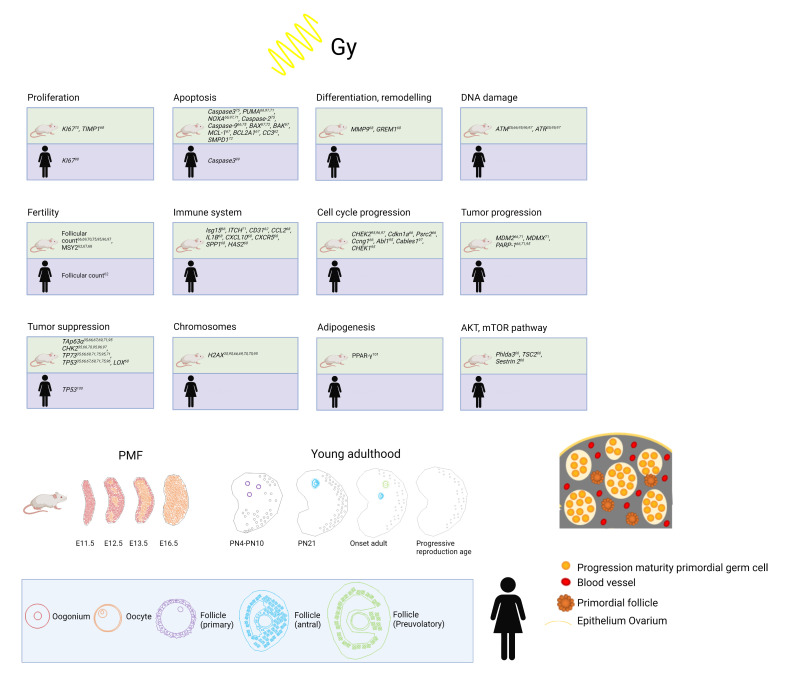
Top: Overview of already established markers concerning ovarian damage including proliferation, apoptosis, differentiation and remodeling, DNA damage, fertility, immune system, cell cycle progression, tumor progression, tumor suppression, chromosomes, adipogenesis, and AKT/mTOR pathway after radiation in human and mouse models. Bottom: Development from PMF pool to young adulthood in mice and humans and human folliculogenesis and detail in PMF pool in humans at birth. Adapted from [54,55,56,57,58,59,60,61,62].

**Table 1 cells-11-01219-t001:** Effect of low doses of radiation during postnatal life on mouse ovaries.

Area	Mouse Strain	Age at Radiation	Time Post-Radiation	Radiation			Ovarian Reserve	Ovarian Follicular Development and Atresia	Stroma	Reference
				Source	Rate Dose	Total Dose				
**Total Body Irradiation (TBI)**	BALB/c	5 d	5 d	γ-ray	141 rad/min	0.1 Gy	Similar primordial follicles	Similar follicular development	ND	[37]
						0.45 Gy	Near to depletion of primordial follicles	Near to depletion of small primary; presence of secondary follicles		
	C57BL/6	5 d	5 d		NI	0.45 Gy	Depletion of primordial follicles; no follicular renewal	Near to depletion of primary follicles; presence of secondary follicles		[38]
	CD1	5 d	6 h		2.387 Gy/min	0.52 Gy	ND	ND		[39]
	NMRI	8 d	2 d	X- or γ-ray	γ-ray: 35.57 mGy/min X-ray: 0.6 Gy/min	0.02 Gy	= Or < primordial follicles	= Growing follicles		[40]
						0.1 Gy	< Primordial follicles			
						0.5 Gy	Depletion of primordial follicles			
	BALB/c	5 w	2 w	γ-ray	2.1 Gy/min for 0.48 min	1 Gy	Near to depletion of primordial follicles	< Primary, secondary, and antral follicles persistence of secondary and antral in 129 > in BALB/c mice	No fibrosis < Spp1	[41]
	129		2 w						No fibrosis = Spp1	
	CD1	6 w	2 w		2.1 Gy/min for 48 s	0.1 Gy	= Primordial follicles	= Follicular development; = Proliferating and apoptotic cells in follicles and corpora lutea	No fibrosis; = ovarian vasculature; = apoptotic and proliferating cells	[42]
						1 Gy	Depletion of primordial follicles	Near to depletion of primary follicles; < secondary and antral follicles; = early antral follicles; > proliferating and = apoptotic cells in follicles and corpora lutea		
			5 w			0.1 Gy	= Primordial follicles	= Follicular development; = proliferating and apoptotic cells in follicles and corpora lutea		
						1 Gy	Depletion of primordial follicles	Near to depletion of total follicles; no proliferating and = apoptotic cells in follicles and corpora lutea		
	NI	6 w	2 w	NI	NI	0.1 Gy	Near to depletion of primordial follicles	< Primary follicle; = pre-antral and antral follicles	ND	[43]
**Targeted Irradiation**	CD1	6 w	2 w	X-ray	0.037 Gy/s for 27 s		< Primordial follicles	< growing follicles; = antral follicles	ND	[44]

Readouts are reported compared with the non-irradiated control group unless otherwise stated. d: days; w: weeks; NI: not informed; ND: not determined; s: seconds; Gy: gray; <: reduction; >: increase; =: no significant differences. In all cases, animals were exposed to radiation. The effects of LDR on the gonads of the mice were investigated, taking into account the aforementioned variables with the aim of preventing harmful effects of radiation on ovarian reserve, follicular development, and atresia of the ovarian stroma.

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
