# Peer review of "Unlaid Eggs: Ovarian Damage after Low-Dose Radiation"

_cells, 2022, doi:10.3390/cells11071219_

Round 1
Reviewer 1 Report
An excellent review paper. I reccommend publication without any changes.
Author Response
We thank the Reviewer for his comment.
Reviewer 2 Report
This is a well-written and timely review summarizing the current state of knowledge regarding ovarian damage after low does radiation. The review is based on a literature search from 1980 to January 2022, and for the most part discusses the pertinent literature. One of the major strengths of this review is that the authors combined discussions on low dose radiation in clinical practice with experimental animal models, and then summarize future directions for practitioner interested in fertility preservation for cancer survivors. Likewise, pulling together all of this information will help inform both clinicians and patients as to the risk of low dose radiation on future ovarian function. Overall, it is an excellent review that is much needed in the field. However, there are some important references that have been overlooked as well as some inaccuracies that will need to be modified.
- Lines 40-42: Does this sentence refer to radiation as the sole form of anti-cancer therapy? Or in combination with chemotherapy? Some mention in the Introduction of the relative risk to the ovaries of patients receiving only low dose radiation (no chemotherapy) should be made; and if there is no compelling information, this is another justification for this review.
- Lines 17-18: delete ‘remains elusive’ since there are some studies on the effects of radiation on the stroma (the authors’ reference 22). Please revise this sentence: “Studies are emerging to evaluate radiation induced damage to the surrounding ovarian tissue, including stromal and immune cells.”
- Line 18: delete “encompass’, replace with “review”.
- Line 26: delete “finally agreed” and replace with “conclude”
- Line 53: References 6 and 7 do not refer to the importance of the interaction between oocytes and stromal cells during the prenatal period. Please replace these references with those that are more appropriate, for example:
Temporal differences in granulosa cell specification in the ovary reflect distinct follicle fates in mice. Mork L, Maatouk DM, McMahon JA, Guo JJ, Zhang P, McMahon AP, Capel B.Biol Reprod. 2012 Feb 14;86(2):37. doi: 10.1095/biolreprod.111.095208
Lineage specification of ovarian theca cells requires multicellular interactions via oocyte and granulosa cells. Liu C, Peng J, Matzuk MM, Yao HH.Nat Commun. 2015 Apr 28;6:6934. doi: 10.1038/ncomms7934.
At the Crossroads of Fate-Somatic Cell Lineage Specification in the Fetal Gonad.
Rotgers E, Jørgensen A, Yao HH.Endocr Rev. 2018 Oct 1;39(5):739-759. doi: 10.1210/er.2018-00010
- Lines 53-55: please include the following publication on single cell RNA seq in human ovaries that identified theca cells, not currently in the list of cell types provided here. It is well-known that follicles acquire theca cells from the secondary stage onward, and that they are an integral part of follicular function:
Single-cell reconstruction of follicular remodeling in the human adult ovary.
Fan X, Bialecka M, Moustakas I, Lam E, Torrens-Juaneda V, Borggreven NV, Trouw L, Louwe LA, Pilgram GSK, Mei H, van der Westerlaken L, Chuva de Sousa Lopes SM.Nat Commun. 2019 Jul 18;10(1):3164. doi: 10.1038/s41467-019-11036-9.
- Lines 79-84: Please add this ‘classic’ reference in the field to the last sentence of the Introduction; since the authors are reviewing the literature, this published review should be mentioned in their review, strange it did not come up in the authors’ search:
The current knowledge on radiosensitivity of ovarian follicle development stages. Adriaens I, Smitz J, Jacquet P. Hum Reprod Update. 2009 May-Jun;15(3):359-77. doi: 10.1093/humupd/dmn063.
- Lines 159-160: The reasons for ‘favoring’ the mouse model should not include similarities with humans with respect to ovarian morphology and function; nonhuman primates fulfill these characteristics, not mice. This statement as written is too general and misses the fact that mice do not have long follicular or luteal phases (do not have ovarian/menstrual cycles similar to women), the arrangement of follicles in the ovaries are somewhat different, mice are litter-bearing species, and follicles can survive in the ovaries for decades wherein mice they are only around for a year or so. Please make this phrase more accurate by saying that mouse ovaries contain similar cohorts of follicles during the continuum of follicular development with similar functions (autocrine, paracrine, endocrine functions; production of a mature oocyte) and then go on to mention the real reasons mice are valuable (genetically modified strains, tools, etc.).
- Lines 246-248: at the end of this paragraph, please add a statement “However, S1P does not protect the ovarian stroma from fibrosis after exposure to high dose radiation; whether S1P can overcome this effect following low dose radiation remains to be studied.” Since there are so few recent references about radiation in the ovary, even though this reference is about high dose radiation, this contrast can be pointed out. This reference also falls within the interval of the literature search.
Sphingosine-1-phosphate and its mimetic FTY720 do not protect against radiation-induced ovarian fibrosis in the nonhuman primate. Amargant F, Manuel SL, Larmore MJ, Johnson BW, Lawson M, Pritchard MT, Zelinski MB, Duncan FE. Biol Reprod. 2021 May 7;104(5):1058-1070. doi: 10.1093/biolre/ioab012.
- Lines 246-249: The authors could also consider including this above reference in their mention of fibrosis here.
- Lines 250-255: Please consider adding the following reference. This is more recent that the authors’ literature review. Although this publication refers to high dose radiation, it illustrates the kinds of immune cells in the ovary (rather than the uterus) that can be affected as a result of radiation.
Evidence of cancer therapy-induced chronic inflammation in the ovary across multiple species: A potential cause of persistent tissue damage and follicle depletion. Du Y, Carranza Z, Luan Y, Busman-Sahay K, Wolf S, Campbell SP, Kim SY, Pejovic T, Estes JD, Zelinski M, Xu J. J Reprod Immunol. 2022 Jan 31;150:103491. doi: 10.1016/j.jri.2022.103491.
- Lines 256-261: DNA damage is important to study. However, few studies that assess DNA damage in the ovary in turn evaluate DNA repair. High dose radiation is expected to elicit DNA damage from which there is no return. But, would the DNA damage seen with low dose radiation also be accompanied by DNA repair? This is an important point to make.
- Just prior to the Conclusion: some discussion about emerging therapies to ‘spare’ ovarian follicles, such as proton radiotherapy should be mentioned here.
Proton Radiotherapy to Preserve Fertility and Endocrine Function: A Translational Investigation. Gross JP, Kim SY, Gondi V, Pankuch M, Wagner S, Grover A, Luan Y, Woodruff TK. Int J Radiat Oncol Biol Phys. 2021 Jan 1;109(1):84-94. doi: 10.1016/j.ijrobp.2020.07.2320.
Reviewer 3 Report
Reiser et al review the literature on the effect of low dosage irradiation for female fertility. They summarize first the clinical effects in human patients and then results from animal studies.
Overall, this manuscript is not appropriate for publication. It ignores large parts of the published literature and is in certain areas simply wrong.
In the part focusing on the effect of radiation on human patients the very interesting investigations of the effect of the Chernobyl accident and the atomic bombs in Japan is not included. For example M. Yeager et al., Science
10.1126/science.abg2365 (2021).
In these investigations it was shown that irradiation does not cause severe effects in the following generations, meaning that oocytes - if they survive - have a high capacity for repair. These are important aspects that should definitely be included.
The authors are simply wrong when they state that "...whereas effects of smaller doses (0.1-1.0Gy) are mainly unknown". There are many publications that have studied the effect of irradiation in the range of 0.1 - 0.5 Gy in mice. Here a few examples: Suh et al., Nature 2006, Livera_etal_Reproduction_2008; Deutsch etal_Cell_2011; Kim_etal_CDD_2019; Rinaldi etal_Genetics_2017; Fester_etal_CDDis_2022 to name just a few.
This manuscript contains only 36 references. I have never seen so far a review with such a low number. A five to tenfold higher number would be appropriate. As examples of reviews in this area I recommend Spears_etal_HR_2019; Woodard_Trends in Cancer 2016 or Gebel etal_Molecules 2020.
Reviewer 4 Report
Thank you for allowing me to review the paper "Unlaid eggs - ovarian damage after low dose radiation", which offers an overview of mouse studies evaluating low dose radiation and the effect thereof on the ovary.
In general, the methodology could be improved. The literature search is described, but the specific inclusion or exclusion criteria are missing. Were papers selected only to report on mice data or were there no studies in humans? What about the age of the mice, were they all representative of childhood cancer? Is the paper actually focussing on childhood cancer or childhood and post-pubertal cancer and RT?
With regards to the description of the studies, the reader expects that the paper clarifies what the studies have shown, what the difference are, and what could explain them. The different strains of mice are not discussed, different ages at radiation, .. etc.
Figure 2 is nice but does not add anything to the paper, as the content is not described in the text. The references for all markers are not listed, the markers are not added to table 1 (assuming the same markers were used in the 5 studies). The relevance of figure 2 should be clarified.
The text describes the 5 studies, but also brings in data from prenatal mice, and then switches to humans and FACS. Maybe the description of the 5 studies can be separated from the other data?
The text is difficult to read due to the lack of explanation of all abbreviations at first use (LDR, VMAT, FACS, .. ) and inconsistency in using the abbreviations. The references are usually numbered, but some are written in full (line 229, 238, 261).
Some minor comments:
Line 43 - a word is missing prior to 'cell death"
LIne 63, please consider the recent review by Hoekman for further data on ovarian transposition (Hoekman et al, Eur J Surg Oncol 2019;45: 1328-1340)
Line 64 - please adapt the sentence reading 'OT may not act as a safe procedure for FP" as this is not in line with the recommendations from ESHRE (https://doi.org/10.1093/hropen/hoaa052), American society of Clinical Oncology (Lee, et al., 2006, Oktay, et al., 2018) and the National comprehensive Cancer Network (Koh, et al.,2019).
Line 66- please remove the brackets with "non" - for girls, one would not preserve embryos
Line 102 - please define IMRT and VMAT at first use
LIne 231 - adapt "figure x"
Line 251- NK cells are mentioned twice
Figure 1 - please list the exclusion criteria for the 9 studies.
